# Aroma Volatiles in Litchi Fruit: A Mini-Review

Zhuoyi Liu [1,2,3], Minglei Zhao [1,2,3,*] and Jianguo Li [1,2,3,*]

1 Guangdong Laboratory for Lingnan Modern Agriculture, Guangzhou 510642, China
2 State Key Laboratory for Conservation and Utilization of Subtropical Agro-Bioresources, South China Agricultural University, Guangzhou 510642, China
3 College of Horticulture, South China Agriculture University, Guangzhou 510642, China
* Correspondence: zml503@scau.edu.cn (M.Z.); jianli@scau.edu.cn (J.L.)

**Abstract:** Aroma is considered a fundamental component of fruit flavor. Variations in the composition and content of volatile organic compounds (VOCs) contribute to noticeable differences in fruit aromas. Litchi is a delicious tropical and subtropical fruit, and a large number of germplasm resources with unique aromas have emerged during the past 2000 years of cultivation. In this review, our aim is to collect, compare, integrate, and summarize the available literature on the profiles of VOCs of 25 litchi cultivars. We showed that a total of 556 VOCs were reported from litchi fruit, and the aroma of litchi is mainly determined from the content and composition of monoterpenoids and alcohols, including linalool, geraniol, limonene, terpinolene, *β*-citronellol, *p*-cymene, nerol, *α*-terpineol, *cis*-rose oxide, *β*-myrcene, 4-terpineol, citral, and neral (*cis*-citral), which might contribute to the rose-like or citrus-like aroma of litchi fruit. Moreover, sulfur-containing volatile compounds (VSCs) possibly impart a special flavor to litchi fruit. This review would be a valuable resource for researchers aiming to improve litchi aroma quality by elucidating the possible mechanisms underlying VOC biosynthesis and their metabolism in litchi fruit.

**Keywords:** *Litchi chinensis* Sonn.; volatile organic compounds; monoterpenoid; alcohols; volatile sulfur compounds





## 1. Introduction

The evaluation of fruit flavor is a multi-faceted and highly complex process, which is mainly derived from the complex interaction between one's own taste, smell, and psychological state and the chemical components in the fruit. Taste receptor cells in the human tongue can sense sour, sweet, bitter, salty, and umami tastes caused by the different combinations of soluble sugars (glucose, fructose, sucrose, etc.), organic acids (malic acid, citric acid etc.), and amino acids (glutamic acid and aspartic acid). The epithelial receptors in the nose can smell the fruit aromas caused by various volatile organic compounds (VOCs) (terpenes, alcohols, esters, ketones, aldehydes, etc.) [1–3].

Aroma generally develops from some low molecular weight VOCs with a boiling point of 50 °C to 260 °C under normal pressure. Aroma is highly regulated by the external environment and can be released from plant roots, stems, leaves, flowers, and fruit tissues. In nature, plant VOCs can attract pollinators to assist in pollination [4,5], drive herbivores away from their predation [6,7], and enhance defenses against pathogen attacks [8,9]. On the other hand, different VOCs result in distinctive plant aromas, which could increase the value of botanical commodities [10,11]. Diverse VOCs create unique and different aroma flavors, such as *α*-ionone, *β*-ionone, and geraniol, which present floral scents; maltol and vanillin, which produce sweet scents; some low molecular weight ester compounds, such as ethyl acetate ester, ethyl butyrate, and butyl acetate, which generate fruity aromas; hexanal, *cis*-2-hexenol, and *cis*-2-hexenal, which present delicate fragrances; benzothiazole, which creates a rubbery aroma; 2-acetyl-2-thiazole, which exhibits a nutty taste; and dimethyl trisulfide, which exhibits a pickle aroma [12–14]. Unlike soluble sugar

and organic acid accumulation, VOCs could be active even at picomolar to nanomolar concentrations, which is enough to alter the fruit aroma [15]. In recent years, VOCs have gained attention because the aroma is one of the most appreciated fruit characteristics in fruit quality. Gong et al. [16] found that $C_9$ alcohols and aldehydes produced by the fatty acid oxygenase pathway are important sources of aroma during the growth and development of watermelon. Among them, the four key $C_9$ aldehyde compounds, including (E,Z)-2,6-nonadienal, *trans*-2-nonenal, 3,6-nonylidene-1-ol, and *cis*-3-nonen-1-ol, contribute to the unique sweet aroma of watermelon. Liu et al. [17] explored apple VOCs and found that the composition of VOCs had huge differences among different apple cultivars. The main VOCs in the "pink lady" and "Fuji" apples were esters and terpenoids, while in the "Ruixue" apple, the main VOCs were aldehydes and terpenoids. After combining with transcriptome analysis, it was found that the difference in the related synthetic genes and transcription factors in the fatty acid, isoleucine, and sesquiterpenoid metabolic pathways among cultivars led to the diversity of VOCs.

Litchi (*Litchi chinensis* Sonn.), a subtropical and tropical fruit, has captured the heart of consumers worldwide due to its unique taste, exotic flavor, appealing fruit color, and high nutritional profile. Litchi is grown in over 20 countries and has a long cultivation history of more than 2000 years [18]. Through the combination of genomics and bioinformatics, Hu et al. [19] found that litchi originated in the Yunnan Province of China. After a thousand years of domestication and evolution, litchi evolved as an important fruit crop. Previously, researchers paid more attention to the color and nutritional value of litchi fruit than the flavor [20,21]. However, the unique aromas of fruit crops are also one of the core criteria for the quality evaluation of fruit. Here, we integrated and summarized the available reports on the volatile profiles of diverse litchi cultivars. This review would provide some directions for exploring the possible mechanisms underlying the volatile biosynthesis and metabolism of litchi aromas.

## 2. The Profiles of VOCs in Litchi Fruit

Over the past four decades, the VOCs in 25 litchi cultivars were detected using different techniques. Here, in order to more clearly illustrate the VOC profiles among these litchi cultivars, we summarized and integrated the results from 20 references available, including the production area, detection method, and total number of VOCs in a specific cultivar.

As shown in Table 1, there were two studies from the United States of America. Johnston et al. [22] first identified 42 VOCs from an unknown cultivar of litchi obtained from the Florida lychee growers association and believed that β-phenethyl alcohol, its derivatives, and terpenoids comprised the major portion of the VOCs in litchi fruit. A total of 31 VOCs were detected in "sweetheart" litchi fruit by Feng et al. [23], among which methional and geraniol exhibited the highest flavor dilution factors (FDs) through aroma extract dilution analysis, indicating that these two VOCs were probably the most important characteristic aromas.

The VOC compositions of 18 Chinese varieties were determined with GC-MS. According to the results from 15 references, the number of VOCs varied greatly, ranging from 22 ("Gualv" (GL)) to 173 ("Huaizhi" (HZ)). Generally, the more studies carried out, the more VOCs that were detected in a specific cultivar. Five studies analyzed the VOCs in HZ and "Heye" (HY). Rose oxide, 1-octen-3-ol, linalool, geranial, and geraniol were considered the main aroma compositions of HZ fruit by Wu et al. [24]. However, this conclusion was different from those of Li et al. [25] and Fan et al. [26], who found that anisyl alcohol, benzyl alcohol, germacrene D, β-myrcene, and methyl benzoate contributed to a large proportion of the aroma of HZ fruit. There were 161 VOCs identified in HY fruit. Lu [27] showed that rose oxide had the highest odorant activity value (OAV) at 1223.3, implying that it might play a dominant role in determining the aroma of HY fruit. Six studies analyzed the VOCs in "Nuomici" (NMC), and a total of 143 VOCs were detected in NMC fruit, among which geraniol, guaiacol, vanillin, 2-acetyl-2-thiazoline, 2-phenylethanol, (Z)-2-nonenal, α-damascenone, 1-octen-3-ol, furaneol, and linalool were found to be the most

odor-active [32]. Interestingly, three studies [29–31] found that the contents of α-selinene and limonene were relatively higher than other VOCs, suggesting that these two compounds also contribute more to the aroma in NMC fruit. "Guiwei" (GW) had 114 VOCs in total, and rose oxide, geraniol, and linalool had high OVA values, which had decisive effects on the aroma [27]. In addition, Xu et al. [28] found that the content of α-pinene was also higher in GW fruit. A total of 111 VOCs were detected in "Baila" (BL), and the relative content of hexanal, (E)-2-hexenal, farnesol, terpinolene, geraniol, and β-citronellol were high in the volatile profiles [24,32,33]. A total of 104 VOCs were detected in "Feizixiao" (FZX), and *cis*-rose oxide and 1-octen-3-ol were the most potent odorants for OAVs [24], and nerol and *cis*-pinane had a higher share in terms of relative content [28,30]. "Yuhebao" (YHB) had 48 VOCs in total; 1-methoxy-2-propanol, acetoin, β-pinene, terpinolene, limonene, and nerol were thought to have an important effect on the aroma due to their relatively higher content [28,30]. "Guanyinlv" (GYL) fruit had 62 VOCs, among which limonene, terpinolene, myrcene, ethyl acetate, prenylacetate, nerol, isoprenol, and 1-octen-3-ol were the primary components in the aroma compositions [35]. "Bingli" (BL2) produced 63 VOCs, among which hexanal, 6-methyl-5-hepten-2-one, limonene, and α-muurolene were the major constituents for the characteristic aroma [36]. "Lingfengnuo" (LFN) had 41 VOCs, and myrcene, 2-carene, and limonene exhibited high relative content [26]. By comparing the detection results with the NIST database and referencing the relevant literature, it was found that "Shuangjianyuhebao" (SJYHB) had 41 VOCs, in which tetradecamethyl cycloheptasiloxane, butyl acrylate, dodecamethyl pentasiloxane, diisobutyl phthalate, and β-caryophyllene might be the main aromatic compounds [29]. Wu et al. [24] detected 42, 41, and 37 VOCs in "Zhengfeng" (ZF), "Xiangli" (XL) and "Jizuili" (JZL), respectively. These three cultivars had high rose oxide aroma activity. Additionally, (E)-2-hexenal, hexanal, and 1-octen-3-ol in ZF also had high OAV values, indicating that ZF fruit presents a more complex aroma than other cultivars. There were a few VOCs in "Wuyejiu" (WYJ), "Lizhiwang" (LZW) and "Gualv" (GL). The WYJ fruit had 34 VOCs in which acetic acid, isoamyl alcohol, and methyl acetate were the most abundant [37]. The LZW fruit had 28 VOCs, out of which ethyl acetate, acetoin, and 1-methoxy-2-propanol were the main constituents [30]. The GL fruit had 22 VOCs, and the relative contents of benzyl acetate, citronellol, and ethyl linoleate were found to be high [28]. Nevertheless, Chyau et al. [38] found that acetoin, geraniol, 3-methyl-2-buten-1-ol, octanoic acid, 2-phenylethanol, *cis*-ocimene, and butyric acid were the major volatile compounds conducive to the aroma of an unknown litchi cultivar.

　　　Five litchi cultivars from South Africa were used for the detection of the volatile aroma profiles. In particular, a combination of different detection methods, including GC-O, GC-MS, GC-PFPD, and GC-FTIR, were applied. Studies showed that the three cultivars "Mauritius", "Brewster" and "Hak Ip" had common 24 odor components, including acetaldehyde, ethanol, ethyl-3-methylbutanoate, diethyl disulfide, 2-methyl thiazole, 1-octen-3-one, *cis*-rose oxide, hexanol, dimethyl trisulfide (DMTS), R-thujone, methional, 2-ethyl hexanol, citronellal, (E)-2-nonenal, linalool, octanol, (E,Z)-2,6-nonadienal, menthol, 2-acetyl-2-thiazoline, (E,E)-2,4-nonadienal, α-damascenone, 2-phenylethanol, α-ionone, and 4-vinylguaiacol [39]. "McLean's red" produced 19 kinds of VOCs, and germacrene D and α-murolene were the two compounds with the highest contents [40]. Frohlich and Schreier [41] found that hexan-1-ol, (E)-hex-2-en-1-ol, oct-1-en-3-ol, β-bisabolol, limonene, *p*-cymene, terpinolene, α-acoradiene, 3-methylbutyl acetate, geranylethyl ether, 2-methylpropan-1-ol, 3-methylbutyl-3-en-1-ol, 3-methylbutyl-2-en-1-ol, citronellol, geraniol, 3-hydroxybutan-2-one, and 3-methylbutan-1-ol were the main VOCs for an unknown litchi cultivar.

**Table 1.** List of the number of VOCs identified with different methods in different cultivars.

| Producing Countries | Cultivars | Methods of Detection | Number of VOCs | References |
|---|---|---|---|---|
| China | "Huaizhi" | GC-MS | 173 | Wu et al. (2009) [24], Li et al. (2010) [25], Xu et al. (2010) [28], Yang et al. (2014) [29], Fan et al. (2017) [26], |
| | "Heye" | GC-MS | 161 | Hao et al. (2007) [30], Shu et al. (2008) [31], Wu et al. (2009) [24], Xu et al. (2010) [28], Lu (2014) [27], |
| | "Nuomici" | GC-MS | 143 | Ong and Acree (1998) [32], Cai et al. (2007) [33], Chen et al. (2009) [34], Wu et al. (2009) [24], Xu et al. (2010) [28], Fan et al. (2017) [26] |
| | "Guiwei" | GC-MS | 114 | Wu et al. (2009) [24], Xu et al. (2010) [28], Lu (2014) [27] |
| | "Baila" | GC-MS | 111 | Hao et al. (2007) [30], Wu et al. (2009) [24], Yang et al. (2014) [29] |
| | "Feizixiao" | GC-MS | 104 | Hao et al. (2007) [30], Wu et al. (2009) [24], Xu et al. (2010) [28] |
| | "Yuhebao" | GC-MS | 48 | Hao et al. (2007) [30], Xu et al. (2010) [28] |
| | "Guanyinlv" | GC-MS | 62 | Ma et al. (2015) [35] |
| | "Bingli" | GC-MS | 63 | Dong et al. (2022) [36] |
| | "Zhengfeng" | GC-MS | 42 | Wu et al. (2009) [24] |
| | "Lingfengnuo" | GC-MS | 41 | Fan et al. (2017) [26] |
| | "Shuangjianyuhebao" | GC-MS | 41 | Yang et al. (2014) [29] |
| | "Xiangli" | GC-MS | 41 | Wu et al. (2009) [24] |
| | "Jizuili" | GC-MS | 37 | Wu et al. (2009) [24] |
| | "Wuyejiu" | GC-MS | 34 | Xing et al. (1995) [37] |
| | "Lizhiwang" | GC-MS | 28 | Hao et al. (2007) [30] |
| | "Gualv" | GC-MS | 22 | Xu et al. (2010) [28] |
| | An unknown cultivar grown in Taiwan | GC-MS | 25 | Chyau et al. (2003) [38] |
| South Africa | "Mauritius" | GC-O/MS/PFPD | 49 | Mahattanatawee et al. (2007) [39], Sivakumar et al. (2008) [40] |
| | "Brewster" | GC-O/MS/PFPD | 38 | Mahattanatawee et al. (2007) [39] |
| | "Hak Ip" | GC-O/MS/PFPD | 36 | Mahattanatawee et al. (2007) [39] |
| | Unknown cultivar grown in South Africa | GC-MS/FTIR | 34 | Frohlich and Schreier (1986) [41] |
| | "McLean's red" | GC-MS | 19 | Sivakumar et al. (2008) [40] |
| America | Unknown cultivar grown in Florida | GC-MS | 42 | Johnston et al. (1980) [22] |
| | "Sweetheart" | GC-O/MS | 31 | Feng et al. (2018) [23] |

GC-MS: Gas Chromatography-Mass Spectrometry; GC-O: Gas Chromatography-Olfactory; GC-PFPD: Gas Chromatography-Pulsed Flame Photometric Detection; GC-FTIR: Capillary Gas Chromatography-Fourier Transform Infrared Spectroscopy.

## 3. Characteristics of VOCs in Litchi Fruit

In this review, the already identified VOCs were integrated and summarized to explore the potential VOCs that may contribute to the unique aroma of litchi. After the detection of the similarities of the same name and synonym, a total of 556 VOCs were obtained in litchi fruit (Table S1). Then, the VOCs detected from at least 13 litchi cultivars (accounting for at least half of the total number of cultivars) were selected for further analysis. According to this criterion, a total of 21 VOCs were obtained. Notably, the distribution of 21 common VOCs was not uniform among the different cultivars, and some VOCs were missing (Figure 1). These VOCs could be classified into two groups: monoterpenes and alcohols (Table 2).

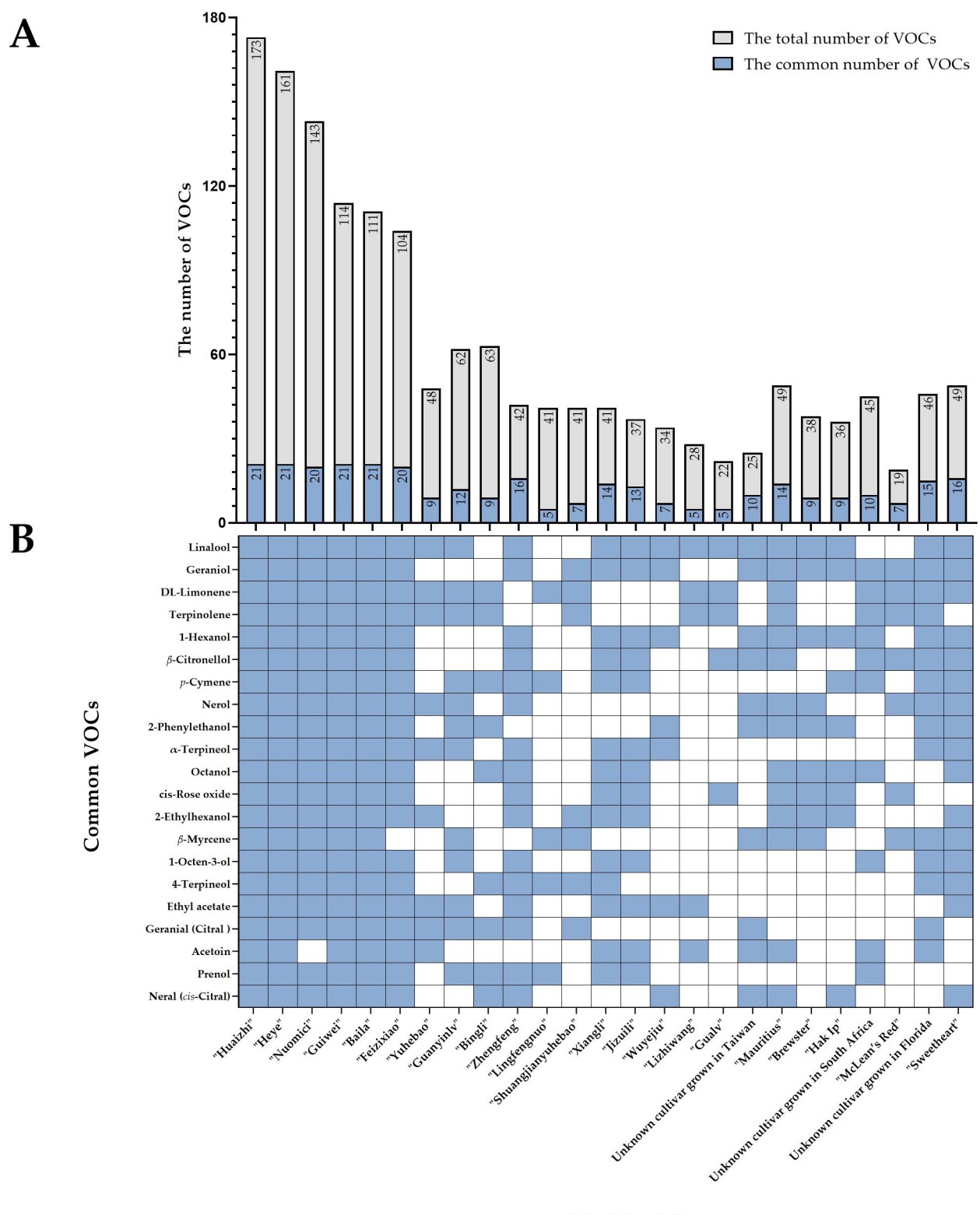

**Figure 1.** The number of VOCs in different litchi cultivars. (**A**) The number of common VOCs in different litchi cultivars. (**B**) The presence and absence of common VOCs in different litchi cultivars. The blue squares indicate the presence of a VOC, and the white squares indicate the absence of a VOC.

**Table 2.** List of 21 VOCs commonly found in litchi fruit and their aroma description.

| CAS | Volatile Organic Compounds | Number of Hits | Scent Descriptions | Substance Classifications |
|---|---|---|---|---|
| 78-70-6 | Linalool | 21 | Floral scent of lilac, lily of the valley and rose | Monoterpenes |
| 106-24-1 | Geraniol | 20 | Rose-like aroma | Monoterpenes |
| 138-86-3 | DL-Limonene | 18 | Orange and lemon-like aroma | Monoterpenes |
| 586-62-9 | Terpinolene | 17 | Lemon, woody and slightly sweet citrus-like aroma | Monoterpenes |
| 111-27-3 | 1-Hexanol | 17 | fruity-like aroma | Alcohols |
| 106-22-9 | β-Citronellol | 17 | Sweet rose-like aroma | Monoterpenes |
| 99-87-6 | p-Cymene | 16 | Aromatic smell | Monoterpenes |
| 106-25-2 | Nerol | 16 | Rose and orange-like aroma | Monoterpenes |
| 60-12-8 | 2-Phenylethanol | 16 | Rose-like aroma | Primary alcohol benzenes |
| 98-55-5 | α-Terpineol | 15 | Clove aroma | Monoterpenes |
| 111-87-5 | Octanol | 15 | Strong oily and citrus aroma | Saturated fatty alcohol |
| 876-17-5 | cis-Rose oxide | 15 | Rose-like aroma | Monoterpenes |
| 104-76-7 | 2-Ethylhexanol | 15 | Special aroma | Primary alcohol |
| 123-35-3 | β-Myrcene | 14 | Light balsamic aroma | Monoterpenes |
| 3391-86-4 | 1-Octen-3-ol | 14 | Mushroom, lavender, rose, and hay aroma | Aliphatic unsaturated alcohol |
| 562-74-3 | 4-Terpineol | 14 | Warm peppery, lighter earthy, and aged wood aroma | Monoterpenes |
| 5392-40-5 | Citral | 14 | Intense lemon scent | Monoterpenes |
| 141-78-6 | Ethyl acetate | 14 | Fruity aroma | Esters |
| 513-86-0 | Acetoin | 13 | Pleasant creamy aroma | Methyl ketone |
| 556-82-1 | Prenol | 13 | Fruity aroma | Alkenyl alcohol |
| 106-26-3 | Neral (cis-Citral) | 13 | Lemon-like aroma | Monoterpenes |

Monoterpenoids are produced by the polymerization of two molecules of isoprene, their oxygenated derivatives with varying degrees of saturation [10,11]. Interestingly, 13 out of 21 VOCs belonged to the monoterpenoid group (Table 2). Litchi fruits are often considered to release a rose or citrus scent, which are thought to be derived from monoterpenoids. It is well known that linalool, citronellol, nerol, and geraniol are the main components of rose flowers and some rose-flavored products [42–44]. In citrus fruit, linalool, octanal, α-pinene, limonene, and (E,E)-2,4-decadienal were considered the key components for the mandarin-like aroma [45]. In litchi fruit, Johnston et al. [22] believed that terpenoids, phenethyl alcohol, and its derivatives might be the main compositions contributing to the floral-like and sweet orange-like aromas. Similarly, Ong and Acree [32] also explained that phenylethyl alcohol, terpenoids, and geraniol were likely the main VOCs providing the floral-like and citrus-like aromas in litchi fruit. Feng et al. [23] described that "sweetheart" had a sweet scent through sensory evaluation, and the sweet flavor obtained the highest score of 5.3 in 11 the flavor evaluations. Combined with the GC-O analysis, geraniol, linalool, and nerol were considered to be linked with a sweet and floral scent. Based on our analysis, it was suggested that linalool, geraniol, β-citronellol, nerol, cis-rose oxide, limonene, terpinolene, and nerol are universal in litchi fruit (Table 2), and the differences in the contents and compositions of these VOCs are important sources of the rose or citrus scents in litchi fruit.

Monoterpenoids are $C_{10}$ compounds derived from the precursor substance geranyl diphosphate (GPP) produced through either the plastidic methylerythritol phosphate pathway or the cytosolic mevalonate pathway. During this process, the transcription factor terpene synthase (TPS) is considered to play an important role in regulating the synthesis of monoterpenoids. For example, CitTPS16 in citrus (*Citrus sinensis* Osbeck) was reported to regulate the synthesis of E-geraniol, which provides unique sweet flavors in citrus fruit [46]. In grapes (*Vitis vinifera*), *VvTPS56* affects the synthesis of aromatic alcohols in the early development stage; *VvTPS52* is involved in the transformation of GPP to produce geraniol, which affects the formation of grape aromas and the quality of winemaking [47,48]. Moreover, studies in apples [49] and kiwifruit [50] also showed that the synthesis of monoterpenoids is closely related to the TPS gene family members. To our knowledge, the relationship between TPS genes and monoterpene synthesis in litchi fruit has not been identified yet. Previously, it was found that terpenes were

the main source of the flower scent, and the terpenoid production in litchi flowers was closely associated with the expression level of TPS gene family members [51–53]. Whether the activity of TPS has an impact on aroma formation in litchi fruit needs further studies.

Alcohols could be derived from multiple biosynthesis pathways, including the oxidation lipoxygenase pathway, terpenoid pathway, methyl-branched pathway, and phenylalanine pathway [54]. As shown in Table 2, 1-hexanol, 2-phenylethanol, octanol, 2-ethylhexanol, 1-octen-3-ol, and prenol were the most common alcohols (occupying 28.6%) in litchi fruit. Among them, 1-hexanol, octanol, and prenol might produce a citrus scent, and 2-phenylethanol and 1-octen-3-ol might provide a rose scent [55–57]. In addition, 2-phenylethanol was reported to be the key VOC in the NMC, "Mauritius", "Brewster", and "Hak Ip" cultivars [32,39], and 1-octen-3-ol was the important VOC in FZX, GYL, ZF, XL, and JZL [24,35]. These findings suggest that the aroma characteristics of alcohols might play an important role in regulating aroma formation in litchi fruit.

## 4. Sulfur-Containing Volatile Compounds (VSCs) in Litchi Fruit

VSCs are a class of sulfur-containing volatile compounds which are able to release intense odors with a very low olfactory threshold that renders it difficult to detect with general gas phase detection methods. Studies in fruits, including strawberries [58], durian [59], pineapple [60], and passion fruit [61], have revealed that VSCs could produce unique aromas.

As shown in Table 3, VSCs were also found to be present in litchi fruit. Johnston et al. [22] found that freshly harvested litchi fruit could release benzothiazole, a rubbery-like VSC, via a mass spectrometer and electrolytic conductivity detector. Subsequently, Mahattanatawee et al. [39] detected eight VSCs in three cultivars ("Mauritius", "Brewster", and "Hak Ip"), including hydrogen sulfide, dimethyl sulfide, diethyl disulfide, 2-acetyl-2-thiazoline, 2-methyl thiazole, 2,4-dithiopentane, DMTS, and methional. These VSCs exude the smell of cabbage scent or garlic aroma. There were differences among these three cultivars in the intensity of garlic or cabbage scents that might be due to the differences in the ratio of these VSCs. Wu et al. [24] detected two VSCs in HY with GC-MS, named 2,4-dithiopentane and 2,3,5-trithiahexane, which imparted the garlic scent in litchi fruit. Interestingly, these two VSCs were only detected in HY grown in Guangxi but not in HY grown in Guangdong, suggesting that the production of 2,4-dithiopentane and 2,3,5-trithiahexane might be dependent on the cultivation environment. Feng et al. [23] also detected two VSCs in "sweetheart" litchi fruit, of which DMTS had a higher flavor dilution factor (FD) of 32, which confirmed that a lower DMTS content could trigger a strong sense of smell perception. Methional is often considered to be an important odorant-active compound in subtropical and tropical fruits [62,63], and it has an extremely high FD factor in the "sweetheart" fruit, reaching an astonishing 1024, suggesting that methional might be the main source of the aroma in "sweetheart" litchi fruit.

**Table 3.** List of VSCs found in litchi fruit and their aroma description.

| Method of Detection | Volatile Sulfur Compounds | Scent Descriptions | References |
|---|---|---|---|
| Electrolytic conductivity detector and mass spectrometer | Benzothiazole | Rubber-like odor | Johnston et al. (1980) [22] |
| GC-PFPD/O | Hydrogen sulfide<br>Dimethyl sulfide<br>Diethyl disulfide<br>2-Acetyl-2-thiazoline<br>2-Methyl thiazole<br>2,4-Dithiopentane<br>Dimethyl trisulfide (DMTS)<br>Methional | Sulfur, fetid<br>Cabbage<br>Moldy, sulfur<br>Dry fruit, nutty<br>Roasted garlic<br>Burning tire, cabbage<br>Cabbage, sulfur<br>Cooked potato | Mahattanatawee et al. (2007) [39] |
| GC-MS | 2,4-Dithiopentane<br>2,3,5-Trithiahexane | Cabbage<br>N/A | Wu et al. (2009) [24] |
| GC-MS/O and AEDA | DMTS<br>Methional | Pickled vegetable<br>Cooked potato | Feng et al. (2018) [23] |

GC-MS: gas chromatography-mass spectrometry; GC-PFPD: gas chromatography-pulsed flame photometric detection; GC-O: gas chromatography-olfactory; AEDA: aroma extract dilution analysis. N/A: no record.

## 5. Future Research Perspectives

With the continuous improvement of people's standards of living, fruits with aromatic odors are more favored by consumers [64–66]. The aromas of litchi fruit are usually described as a rose or citrus scent. However, these descriptions are mostly derived from the direct experience of human olfactory senses, which have strong subjective consciousness and lack the support of scientific data. In addition, VOC detection is affected by the internal and external environment and the isolated method. The VOCs in a specific litchi cultivar might be different under different conditions, which is also a common problem in detecting the VOCs in some vegetables and fruits such as tomatoes [67], peppers [68] and apples [69]. To date, the available studies only focused on the detection and identification of aromatic substances in litchi fruit. Some specific litchi cultivars produce unique aromas. However, the key VOCs that contribute to specific aromas have not been identified yet. Furthermore, the differences in VOC metabolic pathways in different litchi cultivars and their related molecular regulatory mechanisms are still unknown. Therefore, undertaking further efforts are required to fully ascertain the association between the diverse VOCs and the aroma phenotypes, identify the key genes responsible for VOC formation, and unlock its molecular action mechanism.

In addition to the value endowed by humans, fruit aromas also have ecological significance. In nature, the aroma of litchi fruits attracts the *Conopomorpha sinensis* Bradley, a main pest of litchi which inhabits the aril of the fruit and causes a huge economic loss [70]. A previous study found that the antennae of *C. sinensi* could generate odorant-binding proteins (GOBPs), which are capable of binding to different VOCs in litchi fruit. CsGOBP1 could easily bind to the VOCs in GW and FZX, and GsGOBP2 possesses high binding affinity with nine different litchi VOCs, which allows *C. sinensis* to have a selective preference for the aroma types of litchi fruit [71]. Thus, if the mechanisms by which the GOBPs of *C. sinensi* percept the VOCs in litchi fruit are clearly clarified, the application of human intervention to disturb this perception signaling pathway might be an effective alternative strategy to control *C. sinensi* damage to litchi fruits.

In conclusion, we integrated and summarized the available studies regarding the detection of volatile aromas in litchi fruit. It was found that a total of 556 VOCs were detected in 25 litchi cultivars. Among these VOCs, monoterpenes, alcohols, and VSCs were considered the master characteristic volatile compounds. Future in-depth studies should be carried out to investigate the molecular regulatory networks underlying the metabolism of specific VOCs to improve the litchi aroma quality.

**Supplementary Materials:** The following supporting information can be downloaded at: https://www.mdpi.com/article/10.3390/horticulturae8121166/s1, Table S1: Total VOCs in different litchi cultivars.

**Author Contributions:** Z.L. collected the data and wrote the manuscript; Z.L., M.Z. and J.L. were involved in the data discussion and the revision of the manuscript. All authors have read and agreed to the published version of the manuscript.

**Funding:** This research was funded by grants from the Laboratory of Lingnan Modern Agriculture Project (NZ NT2021004) and the Special-funds Project for Rural Revitalization Strategy of Dongguan city, Guangdong province (20211800400032).

**Institutional Review Board Statement:** Not applicable.

**Informed Consent Statement:** Not applicable.

**Data Availability Statement:** Not applicable.

**Acknowledgments:** The authors would like to thank the anonymous reviewers for their comments on this manuscript.

**Conflicts of Interest:** The authors declare no conflict of interest.

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
