# Peer review of "Aroma Volatiles in Litchi Fruit: A Mini-Review"

_horticulturae, doi:10.3390/horticulturae8121166_

Round 1

Reviewer 1 Report

1.           at pag. 2, lines 57-60 - please provide a reference to the phrase;

2.           at pag. 2, lines 84 - please define the FD abbreviation;

3.           at pag. 2, lines 87-88 – please rephrase and improve the statement;

4.           at pag. 2, line 91, please remove: A total of 173 VOCs were detected in HZ. – information presented at lines 88-89;

5.           at pag. 2, lines 92-94 – please rephrase and improve the statement;

6.           at pag. 4, lines 103-106 – – please rephrase; split in two and improve the statement;

7.           at pag. 4, lines 116-122 - please rephrase and split in more phrases;

8.           at pag. 4, lines 123-125 - please provide more information on the siloxane compound founded to the SJYHB sort. What kind of reference compound were used to certify the nature of the siloxanes, in order to eliminate ghosts from column degradation process? The same question in case of butyl acrylate.;

9.           at pag. 5, line 155 - please provide table S1;

10.        at pag. 7, line 234 - please explain what kind of detector is Sulfur in GC-S: Gas Chromatography-Sulfur.

Author Response

Dear reviewer, 

We are truly grateful for your critical comments and thoughtful suggestions. They are definitely valuable and very helpful for revising and improving our paper. We have gone through your comments in detail and revised the article accordingly. In addition, the manuscript had undergone extensive English revision. We hope the new manuscript will meet the requirements for approval. The revised portion in the manuscript is marked in red color. The following are the point-by-point responses to your comments/questions.

Thank you very much for your consideration.

Yours sincerely,

Minglei Zhao

Point 1: At pag. 2, lines 57-60 please provide a reference to the phrase.

Response 1: Thank you for your suggestion. The results of lines 56-60 comes from reference 17 and the modification may be caused by the incoherence of our English expressions. We adjusted the sentence as following:

“Liu et al. [17] explored apple VOCs, and found that the composition of VOCs had huge differences among different apple cultivars. The main VOCs in 'Pink lady' and 'Fuji' apples were esters and terpenoids; while in 'Ruixue' apple the main VOCs were aldehydes and terpenoids. After combined with transcriptome analysis, it was found that difference of related synthetic genes and transcription factors in fatty acid, isoleucine and sesquiterpenoid metabolic pathways among cultivars lead to the diversity of VOCs.”

Point 2: At pag. 2, lines 84 - please define the FD abbreviation.

Response 2: Thanks, FD means flavor dilution factor, we have added as following:

“among which, methional and geraniol exhibited the highest flavor dilution factors (FD) through aroma extract dilution analysis, indicating these two VOCs probably be the most important characteristics aroma.”

Point 3: At pag. 2, lines 87-88 – please rephrase and improve the statement.

Response 3: Thanks, we have adjusted the sentence toThe VOC compositions in 18 Chinese varieties were determined by GC-MS. According to the results from 15 references, ………

Point 4: At pag. 2, line 91, please remove: A total of 173 VOCs were detected in HZ. – information presented at lines 88-89.

Response 4: Thank you for your suggestion, we have deleted it.

Point 5: At pag. 2, lines 92-94 – please rephrase and improve the statement.

Response 5: Thanks, we have adjusted the sentence to1-octen-3-ol, rose oxide, linalool, geranial, and geraniol were considered as the main aroma compositions of HZ fruit by Wu et al. [24].

Point 6: At pag. 4, lines 103-106 – – please rephrase; split in two and improve the statemen.

Response 6: Thanks, we have adjusted the sentence as following:

Six studies analyzed the VOCs in ‘Nuomici’ (NMC) and a total of 143 VOCs were detected in NMC fruit, among which geraniol, guaiacol, vanillin, 2-acetyl-2-thiazoline, 2-phenylethanol, (Z)-2-nonenal, α-damascenone, 1-octen-3-ol, furaneol, and linalool were found to be the most odor-active [28]. Interestingly, three studies [26, 29-30] found that the contents of α-selinene and limonene were relatively higher than other VOCs, suggesting that these two compounds also contribute more to the aroma in NMC fruit. ‘Guiwei’ (GW) had 114 VOCs in total, where rose oxide, geraniol, and linalool had high OVA value which had a decisive effect on aroma [27].         

Point 7: At pag. 4, lines 116-122 - please rephrase and split in more phrases.

Response 7: Thanks, through further verification, we found that ‘Maichi’ and NMC are the same cultivar, so we integrated the data and modified it as follows:

“‘Guanyinlv’ (GYL) fruit had 62 VOCs, among which limonene, terpinolene, myrcene, ethyl acetate, prenylacetate, nerol, isoprenol and 1-octen-3-ol were main in aroma compo-sitions [39]. ‘Bingli’ (BL2) produced 63 VOCs, among which hexanal, 6-methyl-5-hepten-2-one, limonene, and α-muurolene were the major constituents for the characteristic aroma [40].”

Point 8: At pag. 4, lines 123-125 - please provide more information on the siloxane compound founded to the SJYHB sort. What kind of reference compound were used to certify the nature of the siloxanes, in order to eliminate ghosts from column degradation process? The same question in case of butyl acrylate.

Response 8: Thanks, nice suggestions. We have carefully reviewed the literature regarding the VOCs detection in 'SJYHB' fruit. The GC-MS was used to detect the VOCs and the substance properties were determined by comparing with the NIST database and referring to the relevant literature. There were no clear reference substances used to certify the nature of siloxanes and butyl acrylate. The key aromatic substances in ‘SJYHB’ fruit were determined according to the proportion of the relative content of all substances. We have added this information to the manuscript as following (the black font shows the original text, and the red font is the modified content):

 “By comparing the detection results with the NIST database and referencing relevant literature, ‘Shuangjuanyuhebao’ (SJYHB) had 41 VOCs and tetradecamethyl cycloheptasiloxane, butyl acrylate, dodecamethyl pentasiloxane, diisobutyl phthalate, and β-caryophyllene constituted the main aromatic compounds [35].

Point 9: At pag. 5, line 155 - please provide table S1.

Response 9: We have submitted table S1.

Point 10: At pag. 7, line 234 - please explain what kind of detector is Sulfur in GC-S: Gas Chromatography-Sulfur.

Response 10: Thanks, the Gas Chromatography-Pulsed Flame Photometric Detection (GC-PFPD) and the Gas Chromatography-Olfactory (GC-O) were used to detect and verify the sulfur volatiles, respectively. We have made some modifications to table 3 as follows (the black font shows the original text, and the red font is the modified content):

Table 3. List of VSCs found in litchi fruit and their aroma description.

Method of detectiona

Volatile sulfur compounds

Scent descriptions

References

Electrolytic conductivity detector and mass spectrometer

Benzothiazole

Rubber-like odor

Johnston et al. (1980) [22]

GC-PFPD/O

Hydrogen sulfide

Sulfur, fetid

Mahattanatawee et al. (2007) [32]

Dimethyl sulfide

Cabbage

Diethyl disulfide

Moldy, sulfur

2-Acetyl-2-thiazoline

Dry fruit, nutty

2-Methyl thiazole

Roasted garlic

2,4-Dithiopentane

Burning tire, cabbage

Dimethyl trisulfide (DMTS)

Cabbage, sulfur

Methional

Cooked potato

GC-MS

2,4-Dithiopentane

Cabbage

Wu et al. (2009) [24]

2,3,5-Trithiahexane

N/A

GC-MS/O and AEDA

DMTS

Methional

Pickled vegetable

Cooked potato

Feng et al. (2018) [23]

aGC-MS: Gas Chromatography-Mass Spectrometry; GC-PFPD: Gas Chromatography-Pulsed Flame Photometric Detection; GC-O: Gas Chromatography-Olfactory; AEDA: Aroma Extract Dilution Analysis.

N/A: No record.

Reviewer 2 Report

Dear Authors,

I believe that the work is well written starting with the introduction and ending with the perspective studies.

It can be considered that it brings a consistent input regarding the volatile aromatic compounds of lychees.

I have no other observations.

Author Response

Dear reviewer,

Thanks very much for taking your time to review this manuscript and we are truly grateful for your comments and encouragement. We hope so this mini-review can be helpful for the follow-up research on litchi aroma.

Thank you very much for your consideration.

Yours sincerely,

Minglei Zhao

Reviewer 3 Report

Dear Authors, 

Please modify the sentenci in line 87, it is not correct.

The abstract mentions that this review would help researchers elucidate the possible biosynthesis pathways and metabolism, however there isnt really much said about this topic in particular. 

The English is overall good, slight modifications should be made. 

Also mini-reviews would signifficantly benefit from having some figures - it makes it easier for the reader to navigate and undertand the meaning of the short review, however, if not possible, then ok.

Overall the review is good and summarizes the flavours of litchi, The aim is fullfilled in a qualitative manner.

Author Response

Dear reviewer, 

We are truly grateful for your critical comments and thoughtful suggestions. They are definitely valuable and very helpful for revising and improving our paper. We have gone through your comments in detail and revised the article accordingly. In addition, the manuscript had undergone extensive English revision. We hope the new manuscript will meet the requirements for approval. The revised portion in the manuscript is marked in red color. The following are the point-by-point responses to your comments/questions.

Thank you very much for your consideration.

Yours sincerely,

Minglei Zhao

Point 1: Please modify the sentence in line 87, it is not correct.

Response: Thanks, we have adjusted the sentence to The VOC compositions in 18 Chinese varieties were determined by GC-MS. According to the results from 15 references, ………”

Point 2: The abstract mentions that this review would help researchers elucidate the possible biosynthesis pathways and metabolism, however there isn’t really much said about this topic in particular.

Response: Thanks, we agree with you. In this mini-review, we only briefly discussed the synthesis of terpenoids and alcohols, and paid attention to the potential factors responsible for the synthesis of litchi floral fragrance. However, we discussed much regarding the important VOCs in different litchi varieties, which would provide useful directions and ideas for further research to explore the molecular regulatory mechanisms underlying the aroma volatile biosynthesis and metabolism in different litchi fruit.

Point 3: Mini-reviews would significantly benefit from having some figures - it makes it easier for the reader to navigate and understand the meaning of the short review, however, if not possible, then ok.

Response: Response: Thanks, a great idea. We have added a figure showing the number of VOCs (Fig. 1A) and the common VOCs in different litchi cultivars (Fig. 1B) :

Figure 1 The number of VOCs in different litchi cultivars. (A) The number of common VOCs in different litchi cultivars. (B) The presence (blue squares) and absence (white squares) of the common VOCs in different litchi cultivars.

Line 159-161 have adjusted as following:

“Notably, the distribution of 21 common VOCs not uniform among different cultivars and some VOCs are missing (Figure 1). These VOCs can be classified into two groups: monoterpenes and alcohols (Table 2).”

Reviewer 4 Report

Dear Authors,

compliments on the very well-written study.

I have only a few minor suggestions for improving your manuscript.

Line 77: 20 literatures references available, line 87 also.

Line 79: from the United States of America.

Figures are missing. For example, figure representing overlapping findings (most common VOC) from different studies would be beneficial. 

Author Response

Dear reviewer, 

We are truly grateful for your critical comments and thoughtful suggestions. They are definitely valuable and very helpful for revising and improving our paper. We have gone through your comments in detail and revised the article accordingly. In addition, the manuscript had undergone extensive English revision. We hope the new manuscript will meet the requirements for approval. The revised portion in the manuscript is marked in red color. The following are the point-by-point responses to your comments/questions.

Thank you very much for your consideration.

Yours sincerely,

Minglei Zhao

Point 1: Line 77: 20 literatures references available, line 87 also.

Response: Thanks, we did the modifications as your suggestion.

Point 2: Line 79: from the United States of America.

Response: Thank you for your suggestion and we have made some modifications, Details as follows:

“As show in Table 1, there were two studies from the United States of America.”

Point 3: Figures are missing. For example, figure representing overlapping findings (most common VOC) from different studies would be beneficial.

Response: Response: Thanks, a great idea. We have added a figure showing the number of VOCs (Fig. 1A) and the common VOCs in different litchi cultivars (Fig. 1B) :

Figure 1 The number of VOCs in different litchi cultivars. (A) The number of common VOCs in different litchi cultivars. (B) The presence (blue squares) and absence (white squares) of the common VOCs in different litchi cultivars.

Line 159-161 have adjusted as following:

“Notably, the distribution of 21 common VOCs not uniform among different cultivars and some VOCs are missing (Figure 1). These VOCs can be classified into two groups: monoterpenes and alcohols (Table 2).”
